# Towards Systemic Innovation Programmes for Sustainability Transitions: A Comparative Study of Two Design-Led Cases

Svein Gunnar Kjøde

Faculty of Mathematics and Natural Sciences, University of Oslo, 0316 Oslo, Norway; sveingkj@uio.no

**Abstract:** Sustainability Transitions challenge current practices deeply entrenched through vested interests in dominant regimes. In this sense, actors are locked into paradigms that are systemic and resilient to change. In response, opportunities within designerly approaches encompassing systemic innovation's dynamic, multi-stakeholder and interconnected nature are investigated. The adoption of such approaches is evident among progressive actors facilitating systemic collaborations. Consequently, this paper proposes Systemic Innovation Programmes as a concept to define such initiatives, particularly for addressing sustainability transitions. Two contemporary programmes in Norway are presented, and a comparative analysis is made by linking key frameworks from the systemic design and transition to the management literature to clarify their tangency to intentional, sustainable systems change. The study identifies a spectrum of programmatic and faciliatory considerations in practice that broadly aligns with important frameworks from the systems research; however, they are rarely formalised in the programmes' methodology or framing conditions. Thus, the theoretical contribution aims to inform systemic practitioners and policymakers in further integrating sustainable transition perspectives into future systemic change initiatives.

**Keywords:** systemic design; systems change; sustainability transitions; transition management; sustainability-oriented innovation

## 1. Introduction

Grand challenges, such as climate change [1] and environmental degradation [2], are critical, urgent, and systemic in nature. Commonly, it entails that they are complex, interconnected across sectors and actors, and resilient to change [3]. Furthermore, efforts to intervene and direct systems-wide changes are associated with high levels of uncertainty given their long timeframes and inherent complexity and, thus, at risk of unintended and unwanted effects [4]. A growing community of researchers are investigating how such transitional journeys can come about in a just manner that acknowledges planetary and social boundaries [5], defined as sustainability transitions [6–8].

The current research agenda of the STRN [9] affirms an understanding of sustainability transitions as an interconnected, highly collaborative and interdisciplinary endeavour. New technologies and infrastructures are needed, but increasing attention must also be given to the social dynamics, practices, and mental models that constitute the human element of our socio-technical systems [10]. Thus, we must also investigate the organising and facilitation of the processes that influence and create interventions intended for systems change. In response, we observe an interplay between two bodies of research: transition management [11] and systemic design [12]. Several characteristics of systemic processes lend themselves naturally to a designerly approach. The participatory-, interdisciplinary-, and multi-stakeholder aspects of transitioning argue for reflexive facilitation and knowledge brokering that have a long tradition in designerly practices [13–15]. The contribution of design as a catalyst for (business) innovation is also extensively described [16–18], and lastly, studies into the applicability of design in sustainability-oriented, complex contexts are developing [19–21].

This paper explores these links further by studying systemic initiatives defined as Systemic Innovation Programmes (SIP). Such emergence of novel, multi-actor initiatives [22] is essential for systems-level change, as they enable new ways of establishing mandate, legitimacy, dialogic learning, and creating innovations [23]. However, multi-actor collaboration is fraught with issues, such as conflicting agendas and interests, the strategic positionality of business, intellectual property, and similar central factors of current business logic and economic paradigms [24]. Moreover, the longer timelines of systemic innovations frequently conflict with the demand for short-term gains, impacting the fostering of substantial, lasting change. Ultimately, such collaborative efforts in sustainability transitions are experimental with an inherent uncertainty of outcomes, making scoping and agenda-setting challenging [25].

Knowledge still needs to be improved, particularly regarding structural elements that frame the scope and latitude of systemic initiatives. To address this research need, a review of key literature is presented, followed by a case study of two SIP. Thus, this paper aims to inform practitioners and policymakers in the future development of systemic change initiatives in terms of processes-design and framing conditions.

## 2. Theoretical Background

To present the most relevant literature that this study draws from, we organise this section into Systems theory; Sustainability transitions and transitions management; Systemic design and -innovation; and emergent role of design in sustainability transitions (Figure 1). Lastly, an alignment of frameworks is identified to support the case study.

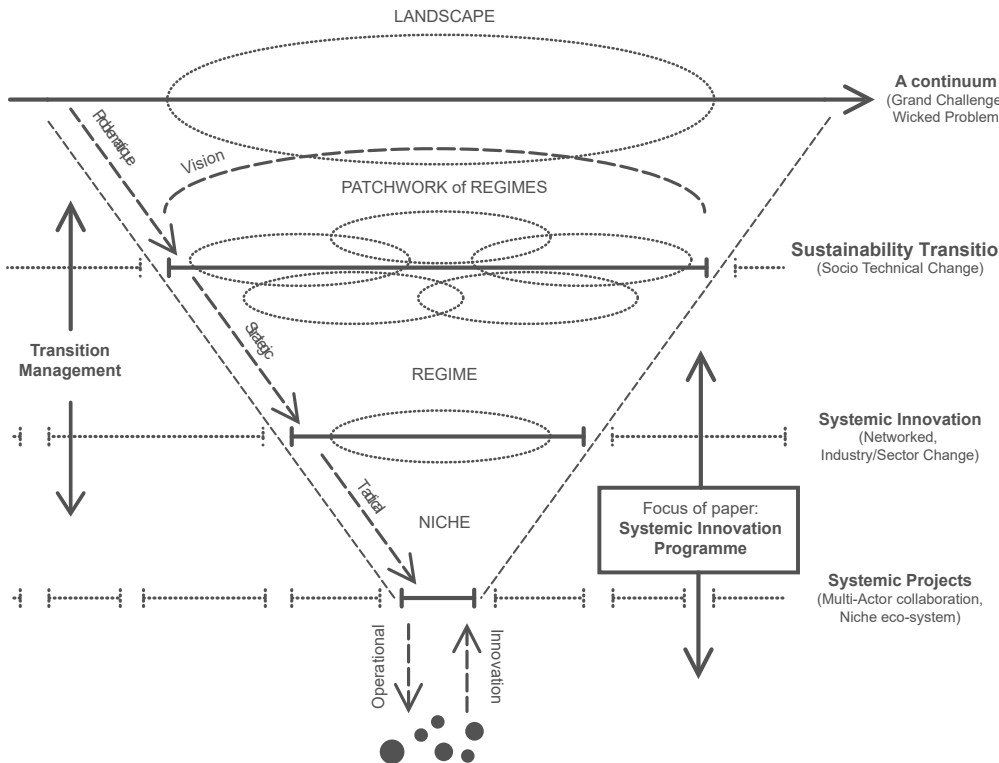

**Figure 1.** Schematic overview of key concepts and orientation of the study. Based on [11,26,27].

### 2.1. Systems Theory

The characteristics of contemporary societal issues suggest that sustainability is a systemic endeavour and must be addressed as such [28]. The complexity that arises from the interconnected, multi-level, multi-stakeholder contexts call for perspectives that can engage with macro, meso and micro perspectives as a dynamic whole. Systems theory and systems thinking [29–31] offer a holistic approach, focusing on synthesis and encouraging exploration of inter-relationships (i.e., context and connections), boundaries (i.e., scope,

scale), and engagement with actors and stakeholders. Such perspectives have increasingly influenced research and practice that engage with systemic issues beyond diagnostics towards systems interventions for sustainability transitions.

The transitioning of societal-scale systems through historical perspectives is thoroughly examined. Notable contributions include the theories of multi-level perspective [32] and socio-technical transitions [26]. The multi-level perspective provides a heuristic that transitions in our built systems are made up of interacting analytical levels—landscape, regime, and niche in which embedded phenomena, including technologies, processes, regulation, knowledge and culture–influence actors' behaviour. Over time, the practices, or "way of doing", of the system will become path dependent through the vested interests of the actors [26] forming a regime. Still, changes to socio-technical systems happen; Their inherent resistance to change is challenged by powerful events (globalisation, digitalisation, climate change) at the landscape level, understood as the larger encompassing context of the regime. Such events can exert enough force on the dominant operational logic of the incumbent regime, presenting "windows of opportunity" to allow the introduction of novel innovations and practices from the niche level to infiltrate the regime, ultimately reconfiguring it by fitness analogous to an evolutionary perspective [33] (p. 4). However, the acute sense of urgency and criticality to current grand challenges has given increased attention to possible ways of accelerating regime shifts through systemic interventions and the intentional governing of such transitions.

### 2.2. Sustainability Transitions and Transition Management

Sustainability transitions [6,34] add a normative, interventionist imperative to systemic change. A notable definition describes it as *"long-term, multi-dimensional, and fundamental transformation processes through which established socio-technical systems shift to more sustainable modes of production and consumption"* [8] (p. 956). Transitioning of socio-technical systems comes with several challenging issues involving rippling effects and change along different dimensions and actors over considerable spans of time [8]. Furthermore, they include changes in social behaviour, practices, and institutional structures. Business models, services, and products are substituted throughout such transitional processes, partly driven by innovations or reconfiguration and complementation of incumbent solutions. The consequent disruption of markets, organisations, technologies, practices, and socio-cultural dynamics can cause considerable harm through unwanted effects and unintended consequences. An obvious tension is thus present as the types of transitions in question are deliberate, purposeful, and normative. To this effect, sustainability transitions incorporate guidance and governance as crucial aspects of the transformation process [34].

The intentional establishing of visionary futures, mass coordination of actors, and strategic orchestration of long-term systemic change strategies have found emergence in the transition management research. Its theoretical roots can be traced from system theories, governance, and strategic (niche) management. An instrumentalist and interventionist approach, it proposes strategic, governing influence on transitions towards more sustainable futures: *The hypothesis underlying transition management is that (collective) understandings of the origin, nature and dynamics of transitions in particular domains will enable actors to better anticipate and adapt to these dynamics so as to influence their speed and direction.* [35] (p. 49). Moreover, it fundamentally acknowledges the need to engage with the networked interdependencies between actors across sectors following the recognition that grand societal challenges are too complex to be solved by any sole actor [36]. Rather than being concerned with policy alone, transition management includes the vantage point of emerging innovations (in technology, business models and practices) at the mesoscale. Leveraging these elements contribute to solutions at the macro-scale, ultimately increasing the potential for course change of current, unsustainable regimes.

### 2.3. Systemic Design and Innovation

The systems and transition theories are increasingly influencing design research, informing new ways to bridge the theory and practice of systemic innovation projects approached by designers. The evolving field of complexity-oriented research, such as systemic- and systems-oriented design, is arguably a response to meet the contemporary challenges presented by clients and society: *"Systemic design can be conceived as optimising processes for group design and decision making under conditions of overwhelming conceptual complexity."* [37] (p. 16).

Designerly approaches have evolved a distinct propensity towards human-centred and constructivist approaches [38]. Thus, they exhibited their efficacy for complex problem-solving [39,40] through their capacity to engage with the dynamics of humans and objects in context. This is evident with deep insight processes latent in design practices that observe systemic behaviours, are deemed critical to identify, and manifest leverage onto systems for lasting change [41]. Thus, design is approached with interest from a systemic innovation perspective, as seen in recent literature [42–44]. More explicitly, Ceschin explored the coupling of design and strategic niche management as intentional reconfiguring towards sustainable product-service systems, emphasising the importance of designers operating strategically to develop effective system innovations [45]. Joore and Brezet identify multiple levels of engagement in the design of products, product-service solutions and, ultimately, systems in a process framework to orientate and coordinate any designed intervention in a multi-level perspective [46]. A common thread in this nascent body of literature is the expansion of design practitioners' role in systems innovation, from a product and service solutions (artefact making) orientation towards facilitation and design of strategy, organisations, and system-change processes. Thus, systemic design is approached with interest from the sustainability perspective, as identified by a recent literature review [21].

### 2.4. The Emergent Role of Design in Sustainability Transitions

Consequently, the emergence of a new role for design becomes evident, which includes the planning and facilitation of systemic innovation in sustainability transitions, even suggested to take the form of human agents mediating transition efforts explicitly [47] (p. 4). A further linking of design and transitioning is explored with socio-technical skills [48], design for sustainability transitions [49], and transition design [50]. With this new role, it becomes imperative that designers gain new operational insight by interfacing with the strategic, governing influence of transition management. However, this would, in turn, require new design processes and skills to be integrated into design-led systemic innovation projects. *"If designers want to play a more effective role in the transition towards sustainability they should be aware of the mechanisms and dynamics that regulate the implementation and diffusion of sustainable radical innovations—and how it is possible to guide and orient them."* [43] (p. 18).

Thus, transition management and systems-related design frameworks, namely *Transition Management Cycle* [11]; *Transition Activities* [51]; *Strategies for Systemic Innovation* [52] and *Systemic Design for Socio-Technical System Innovation* [53], are identified towards a combined framework for evaluating SIP. The author argues that these frameworks are of particular interest for this investigation as they are: (i) rooted in prominent work on socio-technical systems and transition management, (ii) directly integrated into designerly approaches by design scholars, and finally, (iii) all declare a need for further studies into "real-world application". Consequently, their theoretical relations and the resulting combined framework used for evaluating the cases are described in Section 4.2 and the discussion of this paper.

### 2.5. Synthesising the Key Framework for Evaluating SIPs

The following graphical representation in Figure 2 argues for several interconnecting and correlating elements of four key frameworks deemed relevant for studying SIP.

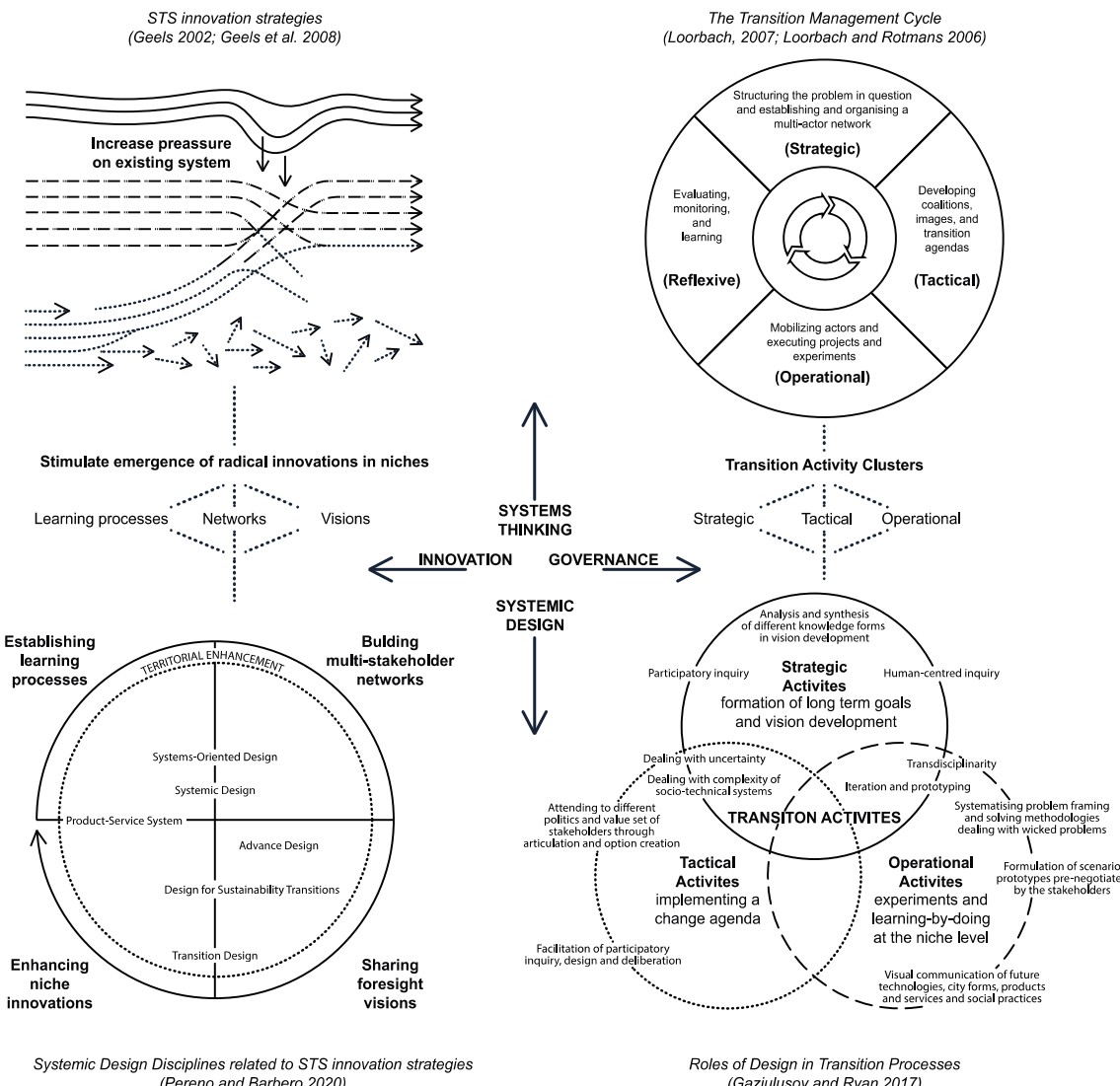

**Figure 2.** Relational overview of key frameworks from socio-technical studies, transition management and correlating frameworks from systems-related design. Based on [11,26,51–54].

The resulting new, combined framework presented in Section 4 aids in structuring the following discussion, functioning as an analytical lens to evaluate the two case studies of this paper. The four key frameworks can further be organised into two complementary sets to clarify their alignment and distinctions:

- Relating to *Transition Management:* Transition Management Cycle [11] and Transition Activities [51].
- Relating to *Systemic Innovation:* Strategies for Systemic Innovation [52] and Systemic Design Tools for Socio-Technical System Innovation [53].

### 2.6. Relating to Transition Management

The *transition management cycle* is a way to organise the so-called systemic instruments [11] (p. 114) in a process with four key phases: (i) Structure the problem in question, develop a long-term sustainability vision and establish and organise the transition arena; (ii) Develop future images, a transition agenda, and derive the necessary transition paths; (iii) Establish and carry out transition experiments and mobilise the resulting transition networks; and (iv) Monitor, evaluate, and learn lessons from the transition experiments and, based on these, make adjustments in the vision, agenda, and coalitions.

The cyclical phases do not represent a strictly sequential process but a need for related activities and outcomes. Furthermore, the transition management research identifies three accompanying activity clusters: *Strategic activities* that involve forming long-term goals and visions. Second, *Tactical activities* are concerned with implementing a transition agenda and connecting actors and activities towards the goal. Finally, *Operational activities* related to learning by doing at the niche level, often through experimentation and disruptive innovations [11] (p. 125). These clusters formed the basis upon which the 12 roles of design, as identified by Gaziuluzoy and Ryan, have been mapped and organised into three categories *in inquiry*, *in process*, and *in outputs*—corresponding to the three activity clusters in transition management. These design roles resulted from observing a 4-year transition project (VP2040), in which only a few were explicit. At the same time, the majority was related to inquiry and process, in alignment with the emerging role of design practitioners as facilitators in complex projects.

### 2.7. Relating to Systemic Innovation

The "*Systemic Design Disciplines for Socio-Technical System innovation strategies*" framework suggests that six systems-related design disciplines can be mapped onto four strategic transition approaches [53]. The Four strategic approaches (or niche-oriented innovation instruments) include (i) Stimulating the emergence and development of radical innovations in niches; (ii) Learning processes: R&D subsidies, subsidies for programmes of experimentation and pilot projects, codification and exchange of experiences, training and competence building, and procurement; (iii) Networks: network management methods, participatory methods to facilitate multi-stakeholder interactions, new platforms, or meeting places, debates, and negotiations, including outsiders or frontrunners; and (iv) Visions: foresight exercises, scenario workshops, and ways of translating long-term visions to short-term actions [52]. Pereno and Barbero continue expanding on the disciplines by identifying 16 related designerly tools deemed applicable in activating the four strategies above:

First, *establishing learning processes* suggests using Holistic Diagnosis and Gigamapping to provide an overview of the complexity involved in systemic innovation. The second strategy, *building multi-stakeholder networks*, focuses on co-creation and stakeholder inclusion through tools like Structured Dialogic Design and Stakeholder Configuration Design. The third strategy, *sharing foresight visions*, seeks to establish collective understanding and strategic vision through the Double-flow scenario method and Multi-level Design Model, among others. Lastly, the fourth strategy of *enhancing green niche innovations* is directed towards the importance of protected spaces for innovations and solutions outside the established regime. As such, the tools address the experimentation, scaling-up and policy aspect of niche innovations.

### 3. Method and Cases

This study investigates the structural organisation and theoretical lineage of two SIP through the perspective of systemic design and transition management. A multi-stage qualitative approach was chosen, including a literature review, framework development, and a comparative case study, as "*qualitative research methods are designed to help researchers understand people and the social and cultural contexts within which they live.*" [55].

The overall process was structured as depicted in Figure 3.

First, the key literature related to (socio-technical) systems theory, sustainability transitions, transition management, and systemic design was identified and reviewed to clarify the notion of SIP. Second, select frameworks from the abovementioned literature were studied and synthesised into a combined framework according to their interrelatedness and complementary aspects for analysing SIP. Subsequently, the two cases of the study were presented, and central aspects of their structure were identified and compared to interpret their academic lineage, methods, practices they infer and outcomes that might be achieved. The main data source was primary documents [56]; That is, content created and made available by the programme owners or partners of the cases and has not been

manipulated by the researcher. This includes publications, websites and media content intended for broad, unfacilitated dissemination. Therefore, it could be argued that the documents can be evaluated as contained, stand-alone 'social facts', which are produced, shared, and used in socially organised ways" [57] (p. 47). A complete list of the documents used for this study can be found in Appendix A. Lastly, a discussion of the two programmes' alignment with the combined framework was conducted to reflect on their efficacy for sustainable transitioning.

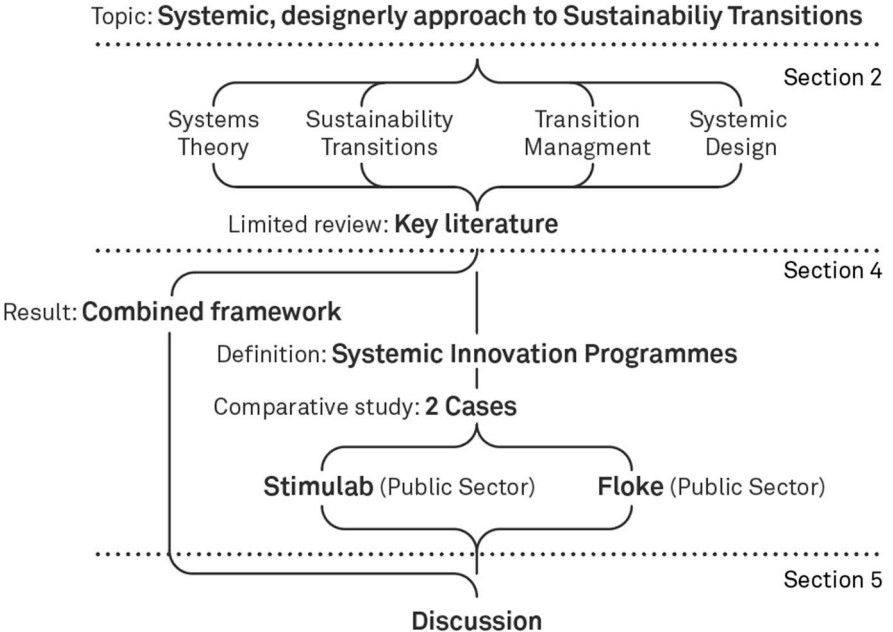

**Figure 3.** Research methods and processes adopted by this study.

It should be noted that the study is part of a broader investigation, beyond this paper, into the emerging role of designers as facilitators of systemic change. That is systemic practices that are situated, social, and relational. The research draws a clear lineage to design practice from the ideas of pragmatism. Dalsgaard has made this link explicit when drawing on pragmatism to prompt a proper understanding of a systemic design situation [58]. This perspective is reflected in the recent development of systemic design, which gravitates toward pragmatism and pluralism [59]. Sevaldson suggests the field itself observes an emergence of generative, adaptive and dynamic design- one in which the so-called real-life context of the challenges drives a primacy of practice; " . . . *if the models do not fit, or they are too cumbersome to operate, they need to be changed.*" [59] (p. 1).

*Two Cases of Systemic Innovation, Stimulab, and Floke*

Due to their specific characteristics, two systemic initiatives in Norway were deemed particularly interesting for investigating the notion of SIP in this study. They are distinctively multi-stakeholder and inclusive processes with ambitions of systems change. Furthermore, they are explicitly design-led, adopting designerly approaches to facilitation.

The StimuLab (Official StimuLab webpages) programme was initiated in 2016 by Design and Architecture Norway (DOGA) and the Norwegian Directorate of Digitalisation (DigDir) on assignment by the Norwegian Ministry of Local Government and Modernisation. It was developed to increase the understanding and utility of human-centred approaches to public sector innovation and has funded 42 projects by 2023. It has identified three key contributions: (i) Facilitate and initiate innovative approaches to public service development, (ii) Connect and utilise existing design-competency in the supplier market, and (iii) Offer a methodology to approach complexity in challenges [60]. The rationale is that the quality of public service offerings will increasingly depend on the ability to

include designerly approaches to innovation that integrate better identification of needs, challenges, and seamless user experiences across channels and touchpoints. It is to be noted that Stimulab is publicly funded through a governmental policy initiative and thus expected to acknowledge ambitions set by principal governmental reports and guiding documents, such as *Digitaliseringsstrategien* and *Innovasjonsmeldingen.* Funnelled through the *Innovative Procurement* format, suppliers are invited to bid on individual projects that DOGA and DigDir manage on behalf of project owners in the public sector. DOGA guides the process, identifying competence and experience as needed, tailored to each project. An interdisciplinary approach is expected, resulting in frequent consortiums of suppliers answering the project calls. This development also reflects the increasing systemic nature of the projects funded in recent years.

The privately held Floke (Website of Floke programme) initiative was coined as a societal innovation programme by its founding organisation, Æra Strategic Innovation, and developed with the conviction that the private sector must take an integral role in addressing the grand challenges of our time. Floke is structured around three distinct core elements: (i) Quadruple Helix collaboration, reflecting that no sustainability challenge can be solved by single actors alone and that collaboration must cross traditional boundaries made by sector and industries; (ii) Open innovation approach that argues for learning and creative processes outside the organisation's internal structures to accelerate radical innovations; and (iii) A designerly approach to strategic business innovation to leverage the participating actors' resources and competencies for engaging with grand challenges by developing an innovative portfolio of synergetic solutions. The programme has initiated 11 projects in several industries and thematic areas, with challenges ranging from sustainable food consumption to circularity in the construction industry. The programme owner and independent domain experts initiate the projects, identifying and investigating grand challenges receiving increased momentum, forming a so-called innovation brief based on insight and knowledge that becomes the call to action. Consequently, relevant actors and key stakeholders are invited as co-funding participants in the main innovation projects, and solutions concepts developed in the project are owned collectively by the participating actors. As a result, numerous ventures have spun out from the programme in the form of new collaborative business ventures, novel product-service solutions, and collaborative strategies.

Both cases presented are total units of analysis with an extensive historical record. However, this study mainly concerns their most current iteration and practices. Thus, the programmatic perspective is supplemented by investigations into recent projects within the programmes to study real-world effects and implications of praxis. As such, they become proxies for recurring phenomena and issues that can support a holistic perspective of programmes beyond their core narratives.

## 4. Results

This section presents the main contributions of this study, including the combined evaluative framework, a further extension of the concept of SIP, and a comparative overview of the Stimulab and Floke programmes.

### 4.1. A Combined Framework for Comparative Analysis

The graphical representation in Figure 4 identifies fundamental relational concepts of the designerly approaches to *transition management* and *strategies for systemic innovation.* A further structuring for this study's investigative purpose is achieved by combining the essential elements into an integrated framework. The framework maps designerly contributions related to levels of engagement and links them to the overarching strategies identified by the multi-level perspective and transition management. This is subsequently used in the discussion section as an analytical lens to structure the investigation into the structural design of SIP and their applicability to sustainability transitions.

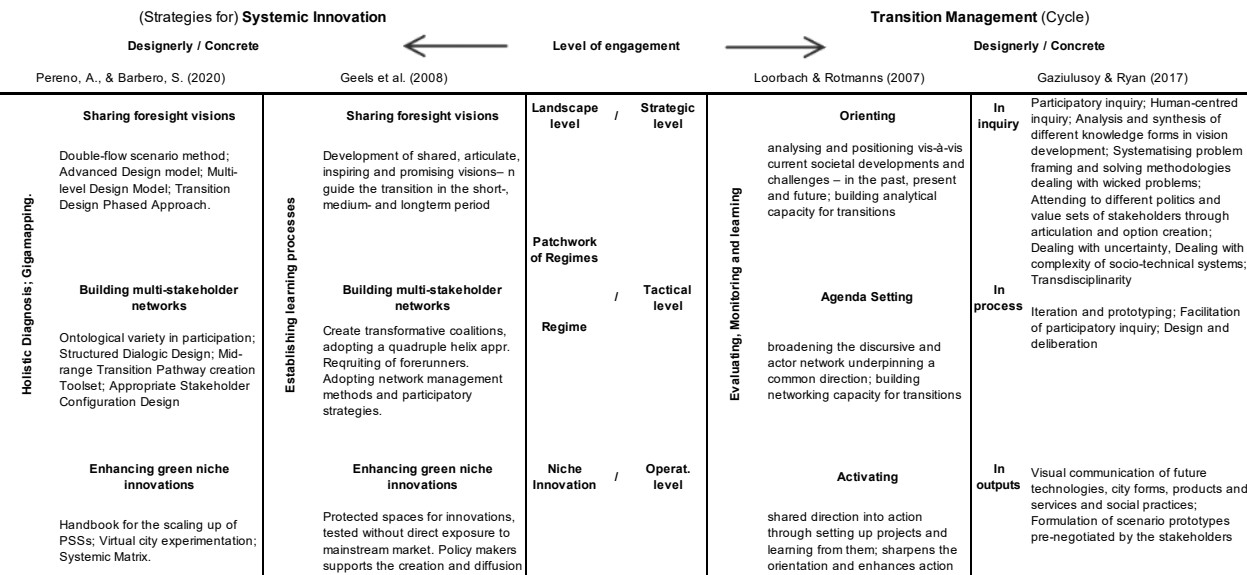

**Figure 4.** A structured overview of the four frameworks shows their interrelated key concepts along a systemic innovation and transition management continuum.

## 4.2. Expanding the Concept of Systemic Innovation Programmes

This study is concerned with the nature of *programmes*. While the term is used interrelatedly with *projects*, several important distinctions of the former's specific structural and managerial characteristics are argued in the literature [61,62]. In management theories, the programme has evolved to encompass and grow beyond wholistic project concepts as a means to engage with challenges that argue for flexibility and benefit realisation beyond the means of the traditional project scope: "*In particular, they suggest that programmes are better at addressing contents and contexts of change characterised by environmental uncertainty and/or ambiguity, complexity, embeddedness and sheer scale*" [60] (p. 235).

A SIP has the ambition of challenging the conditions holding systemic issues in place while acknowledging that such shifts are complex, to the extent that a partitioning into multiple initiatives might be preferable. That is, in managing and coordinating synergistic effect, breaking a grand challenge into sub-projects, addressing different aspects of a composite problem. Furthermore, SIP seeks to frame the initiative in an explicitly experimental narrative, acknowledging the open-aperture approach deemed fundamental for systemic innovation [63]. A further distinction is that SIP focuses on innovation as an interventionist approach, whereas transition management commonly directs attention towards governing influence and policy processes. The SIP cases investigated are also shorter in timespan than suggested in transition management initiatives that could span decades [11] (p. 197). However, definitions are naturally overlapping, as can be seen in an example, which states: "*We, thus, understand systemic innovation and transition programmes as integrated strategies that aim—through coordination structures and activities—to achieve alignment between a set of policy instruments that target different parts of the system.*" [64] (p. 1).

Similar nomenclatures describe novel collaborative endeavours, such as *innovative multi-actor collaborations* [65] or *systems-oriented innovation* [23]. However, SIP is proposed to precisely orientate the study towards the programme as a structural unit of analysis. For the intent of this study, we define SIPs as:

- An open innovation, a responsible initiative [66] that seeks to:
- Organise actors and stakeholders around grand challenges [67].
- Predominantly a multi-actor, cross-sectoral collaboration [68].
- Anticipatory, collective agenda-building for future visions [11] (p. 90).
- A lasting initiative seeking long-term benefit, i.e., learning and evolving capabilities [69], while exploring short-term actions.

- A set of repeatable structural elements (to predict outcomes and minimise risks), such as projects with prescribed forms, boundaries, and tangible deliverables:
- That works synergistically to effect systemic change [62].

### 4.3. Comparative Analysis of Stimulab and Floke

The two cases of this study not only conform with the 7-point definition as described in the previous section but, more importantly, they integrate two additional key elements that are central to the author's ongoing research: (i) They are explicitly design-led in both the facilitatory approach and the promotion of designerly methods and tools. (ii) They have an extensive track record of projects executed in respective programmes, thus amassing significant learnings that have evolved their approaches.

The following structured table (Table 1) identifies and compares their key characteristics relating to designerly approaches to systemic innovation:

**Table 1.** Case overview, including key aspects of the two SIP in question (and theoretical references).

| Programme Aspects | Stimulab | Floke |
|---|---|---|
| Rationale | Stimulating experimental approaches to grow innovation capacities in the public sector and strengthening collaboration with private-sector suppliers. | Sustainability Oriented Innovation through multi-actor cooperation for addressing societal-scale challenges by activating resources and competencies in the private sector. |
| Programme Ownership | Governmental; Norwegian Architecture and Design (DOGA), Directorate for digitalisation (DigDir). | Privately held; Æra Strategic Innovation (Business consultancy company). |
| Beneficiaries | Public sector actors and organisations, general public. | Companies, NGOs, and public sector actors that are affected by the challenge. |
| Key Actors | Multi-stakeholder; Programme owner, project beneficiary, supplier (consultant). | Multi-actor, cross-sectoral Quadruple helix approach. |
| Stakeholder inclusion | Consulted, Co-Design, Ladder of Citizen Participation [70]. | Project owners: Co-design, co-production Eight strategies for co-creation [71]. |
| High level concepts | Mission orientation [72] for creating long-term sustainable, efficient, and quality public services. | Transitioning of industries and sectors towards more sustainable value creation [73]. |
| Systemic perspectives | Wicked Problems [74]. | Wicked Problems [75], Persistent critical challenges [76]. |
| Transition/ Change paradigm | Systems change [41], Niche innovation and portfolio strategy. | Socio-technical change theory [27,77], Developing a portfolio of niche innovations. |
| Sustainability paradigms | Sustainable Development [78]. | Sustainable Development [78], Triple Bottom Line [79]. |

**Table 1.** *Cont.*

| Programme Aspects | Stimulab | Floke |
|---|---|---|
| Innovation paradigm | Public sector innovation [80]. | Open Innovation [81], Shared Value [82]. |
| Designerly Approaches | Human-Centred Design, Service Design, Design Thinking [83]. | Business Design [84], Sensemaking [85,86], Systems Thinking [29]. |
| Designs main role | Designers as creative problem solvers, Design as a cognitive style [87]. | Designers as facilitators and knowledge brokers, Design as an organisational resource [87]. |
| Designerly methods/tools | Participatory design, visualisation, and iterative prototyping. | Co-design, visualisation, and prototyping. |
| Funding | Governmental grant, accessed through procurement processes. | Initial phase is sponsored by programme owner and project partners. Main phases funded by participants (project fee). |
| Project owners | Consultancy winning tender, Single company or consortiums. | Consortium initiated by 2–3 lead actors, Co-financed by all project participants. |
| Project participants | 1–3 Beneficiary actors, 1–3 suppliers -Studio scale [37], | 30–40 participants, 10+ actors, and Arena scale [37]. |
| Project lengths | Annual cycle: 2–5 project started; 5–10 months scoping and onboarding + 6–18 months project run-time. | Annual cycle: 4 projects a year; 5–6 months scoping and onboarding + 6–8 months project run-time. |
| Challenge scoping (project scope) | The overarching theme is established. Additionally, calls for applications are opened. Evaluation of incoming applications by any public actor. -> Resulting in a public procurement call, open for qualified suppliers. | Identification of persistent, critical challenges with a societal interest and actuality. The topic chosen and researched for background -> Resulting in a call to action, recruiting actors and stakeholders. |
| Selection criteria | User-focus and identified user-needs, Clear innovation potential, Multi-actor, cross-disciplinary, Generalisable learning and knowledge, | Societal actuality and momentum, Collective agreement on innovation scope, Cross-sectoral interest and participation, Sufficient value chain representation of actors. |
| Process/ Methodology | Stimulab "Triple Diamond", 3-stage process: Divergent-Convergent thinking/CPS [88], Double Diamond process [89]. | "Floke approach", innovation process: Divergent-Convergent thinking [88], Double Diamond process [89], Three-box solution [90]. |
| Process phases | Three cycles (six phases): 0. Understand/Onboarding/Commitment 1. Diagnosis, 2. Explore + Define, 3. Develop + Deliver. | Six phases, two cycles: 0. Pre-project 1. New Insights, 2. New Ideas +3. Concepts, 4. Experiments, 5. Realisation. |
| Evaluation and Control mechanisms | Dialogue meetings (supplier, beneficiary, program owner), Brief update (problem-/solution scope). | Foundational knowledge report/ Innovation Brief, Core meetings (Facilitator/programme owners + project partners). |

### 4.4. General Consideration of the Cases Relating to Transferability and Applicability

It should be emphasised that the type of ambitious cross-sectoral, multi-actor systemic initiatives represented in this study are currently a rare breed. Furthermore, the two case examples are from Norway and cannot cover a diversity of working-context, such as perspectives from the global south, developing countries, and rural areas. These scope limitations must be considered for further research, such as the possible development of generalisable characteristics towards typology or even theory-building. In this sense, the cases were chosen more in the function as a set of critical issues, constituting a powerful example [91] (p. 20) with the intent to understand "how" and "why" [92].

## 5. Discussion

The following section discusses central attributes of systemic, multi-actor innovation in systemic programmes. First, select, generalisable aspects are covered before the two cases are addressed in a structured investigation through the application of the combined framework, including (i) *Establishing of learning, monitoring and evaluation perspectives* and the three levels of analysis; (ii) *the Landscape/Strategic level*, (iii) *the Regime/Tactical level*, and (iv) the *Niche/Operational level*.

### 5.1. Project Initiation and Framing from the Systemic Programmes

The programmatic approach is beneficial in engaging with systemic issues, as compound challenges are addressed in manageable terms by deploying multiple, coordinated, and synergetic projects. While not explicitly the intention at inception, Stimulab and Floke have evolved from project-driven, social-innovation initiatives towards increasingly systemic, thematically driven programmes. A 2021 Stimulab report identifies an increase in complex, multi-actor projects in the programme. It further states that user-oriented and service design is no longer sufficient to address recent projects' complexity [60]. Thus, it argues for including systemic design to support engagement across sectors, disciplines, and institutions. One supplier identifying a recent Stimulab tender as a 'transformation' project references transition management [11] and Mission approaches [72] as appropriate perspectives. Similarly, a distinct course change of Floke is observed in the shift away from a consumer and end-beneficiary orientation of early initiatives. While tackling a core consumer aspect of unsustainability in current systems, it became apparent that engagement with structural issues, such as business logic, infrastructure and actor dynamics, was critically limited. In addition, in specific domains, such as the construction industry, the interconnectedness of systems is apparent. Responding to these influences, Floke is now distinctly programmatic in its approach in that grand challenges addressed are spun out as multiple, strategically aligned projects. A similar evolution in thematic orientation is observed through Stimulab's adoption of the so-called *Livshendelser* concept, which could be translated to 'Life-events'. Such framing was a substantial step beyond connected innovation towards a more systemic perspective on public service offerings and digitalisation innovation.

### 5.2. Process and Methodological Rationale

Both programmes adopt an overarching approach of phased, divergent-convergent [88] innovation processes. Its origins are attributed to Banathy's seminal work on systems innovation [13], with additional links to Osborn and Parnes with their *Five Diamond CPS model* [93]. The ubiquitous *Double Diamond* process model [89] is a more recent expression of divergent-convergent thinking. However, scholars argue several inherent shortcomings of the model, critiquing the over-simplification of innovation processes in complex environments. The linear representation could also be misinterpreted as a sequential gateway model commonly found in output-oriented processes, i.e., design and engineering [63]. It should be noted that the Double Diamond has seen multiple revisions, addressing some of the critiques and expanding and adjusting the original framework to include systemic- and complexity aspects. The Stimulab programme argues for an additional third sequence at the project start. An extensive diagnostic phase to orient the problem statement towards real user needs, reducing the risk of addressing symptoms and not root causes. Suppliers' project proposals must commit to this *Triple Diamond* process of the Stimulab programme. Such an obligatory, pre-defined alignment is helpful in risk reduction and a means to qualify the methodological competence of suppliers. However, experiences and feedback in the programme have identified the risk of unfortunate dynamics in which such requirements are met in a prescriptive manner at the bidding stage for legal reasons. While in practice, different approaches are implemented by the suppliers.

The Floke programme was conceived as an open innovation, multi-actor process. The innovation approach would enable organisations across sectors to co-develop new business

and value-generating opportunities in addressing societal challenges [82,94]. Its divergent-convergent approach directs attention to macro-drivers as a source for insights that can inform innovation concepts co-designed in collaborative efforts by the participants. As such, it combines key elements from *sensemaking* [85,86], sustainability-oriented innovation [23], and *business modelling* [84,95] in a designerly innovation process.

### 5.3. Stakeholder Identification and Inclusion

The importance of multi-stakeholder inclusion and alignment in systemic change is well argued from systemic innovation and transition perspectives [11,66,96,97]. Such considerations are reflected in cases with both programmes explicitly arguing the need for participatory processes, including stakeholders across traditional sectoral and organisational boundaries. However, the Floke process proposes a highly ambitious approach to stakeholder inclusion: adopting a *Quadruple Helix framework* [68] for identifying actors and stakeholders as participants in advanced co-design activities. A Floke pre-project is initiated to investigate and scope a challenge with an independent knowledge partner (*Academia*) that, in turn, becomes a means to map out relevant actors (*Businesses*), governing institutions and policymakers (*Government*) and civil organisations/NGOs (*Society*). The resulting Floke project may consist of up to 25–30 participants representing a diverse set of perspectives, distributed along a value chain or networked complex, and as such, aligns closely with arguments made in the literature on SOI in the socio-technical perspective [45].

Less attention is present in the programmes to the boundary discussion, and little is formalised in the process or method for stakeholder selection regarding risks, biases, and blind spots. This is regularly addressed in the Systemic Design literature, such as variety deficit, which could lead to critical weaknesses in scoping [37] (p. 26). The Stimulab process can address such issues by re-scoping when concluding the initial diagnostic phase. Furthermore, continuous user and stakeholder involvement are explicitly argued as fundamental to the Stimulab approach. The Floke process is equally ambitious; its co-production approach is considered the most integrated and demanding participatory form, going beyond the co-design in that participants are expected to be heavily involved in creating content and outputs for experiments and solutions [70].

### 5.4. Establishing Learning Processes, Evaluating, and Monitoring

Continuous monitoring and learning are central to supporting sustainability transitions, albeit highly challenging in such a complex environment. Monitoring would include oversight of the overall progress of a systemic programme and its projects and the dynamic development of niche-level and macro events that would emerge within the extended time frames of transition initiatives [11] (p. 17). In addition, monitoring of the systemic programme itself must be considered; This includes network activities and behaviour of actors, but also a continuous evaluation of actions, goals, and interventions that might have been agreed upon. Moreover, an additional layer of complexity is added, as any systemic intervention in the socio-technical perspective will infer learning about changes to social behaviour and models [53], including that of organisations and institutions and individuals. Thus, transition management and systemic innovation strategies argue for integrated, explicit, and formalised learning processes emphasising *"Learning-by-doing and doing-by-learning"* [11] (p. 81).

Both Floke and Stimulab show evidence of extensive revisions to both programme structure and methodological approach from 5+ years of evaluated experience. However, a fundamental differentiating factor between the two regarding learning is how process facilitators are integrated. In Floke projects, facilitators are recruited within the parent organisation by internal selection criteria (domain competence, availability, and experience). Thus, learning cycles and adaptability of the methodology can be both rapid and reflexive [98]. Furthermore, learnings are easily captured within the organisation, as most employees have experience with the programme. However, no integrated routine was found to structure and disseminate such knowledge besides ad hoc de-briefs and spo-

radic workshops evaluating the programme. Stimulab, on the other hand, is subject to public procurement processes, in which suppliers can answer a two-stage competitive bid for each project call. The competing suppliers propose teams and process plans in alignment with the project description. As discussed earlier, such prescriptive formats may impair the suppliers' motivation to explore novel and experimental approaches [99,100] and omit real concerns drawn from previous knowledge and experience from earlier Stimulab projects. Thus, the programme must be vigilant in mediating such risks by gathering learnings beyond the programme organisation from suppliers and project owners, fostering transparency and openness.

The nature of public funding makes dissemination of learnings integral to Stimulab, as it is expected to generate transferable value to public-sector actors at large. Thus, learning processes are, to a significant degree, formalised and structured. Extensive Stimulab programme reports have been published, collating key learning spanning several years and projects, leading to revised project conditions and frameworks.

Revisits of initial scope documents as an evaluative, risk-reduction measure are present in both programmes. In Stimulab, the output of the projects is redefined based on insights from the diagnostic phase. This phase concludes with revising the problem statement, allowing the project to revisit hypotheses and re-scope the project and its priorities. The innovation brief of Floke pre-projects includes a preliminary scope of what could be described as *innovation intent*. This document then acts as an onboarding mechanism for actors and stakeholders, providing a shared knowledge platform that is revised and aligned with participants at the main project start. However, in practice, such knowledge and learning platforms are rarely revisited and revised collectively throughout the projects, contrasting with their suggested importance in the literature [11] (p. 98), [50] (p. 11).

### 5.5. Addressing the Landscape/Strategic Level

The socio-technical theories' landscape level (macro) relates to the larger environment where the transitions occur. Hence, it involves phenomena and issues on a societal/global scale, i.e., climate change, globalisation, international agreements, deep cultural patterns, and macroeconomics [26]. Landscape changes are predominantly slow and long-term, including the factors that may enable them. Moreover, they are not subject to change by individual actors in the short term. Therefore, engagement with this level is centred around orientation, long-term visioning, and strategising in the transition literature and the SIPs. These foci are reflected by systemic design perspectives included in the combined framework. As grand challenges exist on a continuum (Figure 1), scoping of systemic intervention must orient itself to the current landscape's macro trends. Gaziulusoy and Ryan identify several roles of design in such strategic orientation, including "*Systematising problem framing*" and "*Solving methodologies dealing with wicked problems*" [49] (p. 1305). Accordingly, Pereno and Barbero's framework suggest that design tools, such as *double-flow scenario methods* and *MLP design model*, are applicable to "*collectively identify problems, build alternative visions and establish the strategies required to implement them*" [51] (p. 125). However, such phased organisation of design tools may be contrived, as *Gigamapping* [101] would also be helpful for strategic landscape orientation purposes.

Generally, one observes numerous designerly approaches by the Stimulab and Floke facilitators for engaging with the macro level. Visual mapping is extensively used for both orientation and diagnostic purposes. Commonly facilitated as collaborative processes, their primary purpose is identifying the interconnectedness of issues and opportunities within the systemic challenge [101] (p. 250). These approaches are commonly conducted on physical print formats and allow dialogic exploration, including multiple stakeholders with little training. Furthermore, broad investigations into trends and macro-drivers are conducted as intermediary steps towards scenario generation. Such "*analysis and synthesis of different knowledge forms . . .* " [49] (p. 1305) contribute vital insight to the project group when engaging with subsequent future processes. Hence, such explorations include socio-political developments distinctively present in Stimulab, where projects are anchored

in overarching policy processes that concede current governmental white papers and ministerial implementation strategies.

Visioning and futuring [102] are instrumental in establishing systems-level ambitions. In Floke, such vision narratives are initially synthesised from expert interviews and scientific contributions by the knowledge partner, from which an innovation brief is developed. It acts as a call to action for actors and stakeholders that, in turn, collectively engage with futuring processes in the subsequent innovation project. In the Stimulab, a Missions approach is expected as a central conceptual foundation for the projects. Therefore, a mutual narrative is vital in aligning the stakeholders' visions and ambitions. Such futuring (foresight) is closely aligned with the Missions approach through anticipatory innovation [72] (p. 807).

*5.6. Addressing the Regime/Tactical Level*

The particular attention directed towards multi-actor networks in transition management reflects the multi-level perspective of socio-technical systems. It involves the formation and persistence of regimes [26]. The ample literature points to the importance of such networks and collaborative environments in addressing the systemic issues in sustainability challenges, [22] an approach deemed fitting with the interconnectedness of regimes (a patchwork of regimes) constituting several societal systems. The complex dependencies and lock-in of regimes are at the core of their resistance to change. As such, Stimulab and Floke are expected to construct their rationale around multi-stakeholder, cross-sectoral collaboration. In the Floke programme, incumbents and innovators (start-ups and outsiders) are expected to form new project partnerships across organisational boundaries as opportunities for shared value [82]. The potentiality of such partnerships is richly described [22] (p. 78); nonetheless, Floke emphasises the provisioning of a collaborative, safe environment for experimentation, often lacking at a regime level. As a result, innovation ecosystems extend beyond the traditional partnership that may be limited to the expected flexibility and adaptability needed in systemic innovation.

The strategic recruiting of actors identified from value chain systems in Floke is analogous to *transformative coalitions* and argues that actors could also be incentivised to help address collective challenges within industries. Described as "*partnerships of multiple actors that generate innovation through knowledge flows*", Pereno and Barbero argue that such coalitions to be supported by designerly methods and tools developed for "*active engagement of multiple stakeholders*" and include stakeholder selection and interaction maps, structured dialogic-design, and transition pathways creation [51] (p. 123). Such tools seek to encompass the dynamics of dialogic processes between actors and their interests in generating requisite variations of perspectives and innovations to address systemic challenges [103]. They are central to the Floke process and included as collaborative worksheets in facilitated workshops. An equivalent role of design is identified at the tactical *in-process* level by Gaziuluzoy and Ryan, which describes the "*facilitation of participatory inquiry, design and deliberation*" [49] (p. 1305).

Regime engagement is less integrated at the project level in the Stimulab programme, which increases tensions in boundary discussions within individual projects. However, the ambition of coherent and seamless public services in the life-events framework has drawn projects together for ad hoc knowledge exchange. Additionally, the recently adopted Missions approach challenges traditional practices in governing, organising, and managing public sector services and explicitly suggests that the output of such projects are not only solutions but portfolios of initiatives that synergistically address the more extensive, ambitious challenge in the mission statement.

*5.7. Addressing the Niche/Operational Level*

The concept of niche level is central to the socio-technical theory. It is described as "-*where radical innovations can emerge, and new concepts can be tested in a protected environment*" [50] (p. 8). Niche innovations and, similarly, transition experiments in transition management

are seeds of solutions, practices, and structures that could ultimately emerge relevant at the regime level, replacing and reconfiguring unwanted or unfit existing ones: "*Transition experiments are iconic projects with a high level of risk that can make a potentially large innovative contribution to a transition process.*" [25] (p. 176). Experimentation is thus fundamental to niche-innovation development, and critically so for market and user acceptance, increasing their likelihood to reach (commercial) scale and regime integration. The intentional germinating of niche innovation is thus subject to intentional steering in sustainability transitions to develop numerous synergetic transitional experiments and solutions or portfolios [96,104].

The innovation portfolio is central to Stimulab and Floke's programme rationale. However, two critical distinctions between the two can be found in the outputs at the project level; in Stimulab, the overarching system orientation resides with the Mission [72] (p. 805), while each project's ambition is to experiment with the solution level, with users and stakeholders. Nevertheless, a strong service-design orientation risks the output-agnosticism central to systemic challenges, as suitable solutions might call for other types of interventions or even at different parts of the systems.

In the multi-actor Floke, all projects develop portfolios; even so, strategies to coordinate and synergise individual portfolios around thematic areas or domains reside centrally in the programme organisation. Individual solution concepts are not bound to participating actors; instead, the portfolio acts as a mutual repository for the group to be engaged with, depending on individual interests, resources, strategic relevance, and similar. In Stimulab, on the other hand, innovations are inherently linked to the project owners at the onset. Thus, the programme organisation performs a central role in disseminating learning and synergising effects across the project portfolios in alignment with their mission perspective. Missions and systemic innovation share key characteristics, such as being vision-led and arguing for a portfolio of interventions. However, the nature of such innovation ecosystems (analogous to strategies for enhancing niche innovations) makes goal-setting and evaluation challenging for any systemic programme.

Floke projects are inclined towards sustainable business model innovation [105,106]. As such, it will be subject to high levels of uncertainty and challenging feedback lag in the time scope of the innovation process. Therefore, the participating actors focus experimentation around issues of value proposal and market-fit [84] at the strategic level, arguing for the designerly approach to sustainable business modelling [17,107]. The outcome being business model concepts, the portfolio's primary purpose is thus oriented towards partnership formation and decision-making in the participating organisations.

## 6. Conclusions

This study has investigated two contemporary systemic programmes, Stimulab and Floke, with decidedly designerly approaches to systemic innovation. A combined framework was developed, integrating key perspectives from transition management, socio-technical innovation theory and systemic design, and subsequently used as a lens to structure an inquiry into the efficacy of the programmes concerning sustainability transitions.

The framing and structure of the programmes reflect overall systemic considerations in the combined framework in a credible manner. Furthermore, designerly approaches to systemic innovation are increasingly present and integrated into the framing of projects- mirroring recent research into the role of design in such contexts. However, these advanced systemic perspectives are recent additions to the programmes, and long-term effects are to be evaluated. Several challenges at the programme level were identified in the discussion.

The compound problematic of grand societal challenges makes boundary discussion increasingly demanding for the programmes regarding project strategies and initiation. Such challenges arguably exist on long-time scales (continuums), with a consequent need for establishing continuous learning and accumulation of thematic knowledge. The relatively short project processes within the programmes (6–12 months) compound the issue as

suppliers, actors and stakeholders are frequently onboarded and egress. The question arises whether such systemic insight should reside within the individual projects and suppliers, as with Stimulab, or be readily accessible throughout the innovation ecosystems of the programmes.

Intentional systemic change relies on a synergetic portfolio of interventions by diverse actors and stakeholders. Since transition experiments are frequently high-risk and resource-demanding, dissemination of learning becomes critical at the niche innovation level that constitutes the projects' main output [25] (p. 176). The substantially delayed observable effects of niche innovations at the meso and macro levels further argue for establishing innovation ecosystems as part of the programme rationale. This suggests that the programmes must address policy and regulatory ecologies to enable a protected or experimental space for the portfolios to develop and scale [50] (p. ii). This experimental nature of the programmes challenges traditional forms of collaboration and funding, revealing a tension between the need for open-ended processes and effect orientation. The open aperture approach of systemic innovation is adopted as experience shows that root causes might need to be better understood for larger complexes of challenges at project onset, which puts increasing pressure on the programmes to reliably onboard and manage expectations. Additionally- the notion of prescriptive reproduction of innovation becomes incompatible with current systems perspectives.

Finally, the inherent challenge in communicating system transition theories and strategic management is evident at several levels. Boundary discussions and project framing become challenging for programme owners and project participants. They must increasingly be understood as a recurring, re-scoping activity in system innovation. Furthermore, experiences show that the transfer of project portfolios becomes challenging, partly due to the loss of learnings generated within the project processes and the complexity of implicit organisational changes necessary for implementation.

This study does not contend that designers should practice transition management per se in Systemic Innovation Programmes but rather an investigation into how current systemic design practices may or may not readily interface with transition processes. Facilitators in such systemic programmes will likely be required to align and acknowledge governing influences of transition management in the future. In such a context, it will become imperative that designers understand how such perspectives could be integrated and nurture the essential feedback loops necessary for sustainability transitions.

**Funding:** This research received no external funding.

**Informed Consent Statement:** Not applicable.

**Data Availability Statement:** Not applicable.

**Conflicts of Interest:** The authors declare no conflict of interest.

**Appendix A**

| | ID | Document type | Title / Name | Description | Publication/ retrieval date |
|---|---|---|---|---|---|
| Stimulab Innovation Programme | #01 | Print publication | *StimuLab-Brukerorientert offentlig innovasjon – råd og erfaringer fra frontlinjen* | Programme report, summary of learnings | 2021-04-22 / 2021-09-13 |
| | #02 | Web page | *Veiledende bilag til SSA-O – Oppdragsavtalen* | Main document including background, description and scope | 2021-04-22 / 2021-09-13 |
| | #03 | Digital document | *Konkurransegrunnlag Fase 2 - P 07-20 Starte og drive en frivillig organisasjon* | Conditions for the Stimulab tender | 2021-04-22 / 2021-09-13 |
| | #04 | Digital document | *Bilag 2 Konsulentens spesifikasjon av Oppdraget* | Offer from supplier, including approach | 2021-04-27 / 2021-09-13 |
| | #05 | Digital document | *Notat vedlagt endringsbilag* | Change-document, supplier project brief | 2018-11-26 / 2021-10 |
| | #06 | Print publication | *Endelig rapport - Starte og Drive en frivillig org* | Final deliverable, project and process report | 2018-11-26 / 2021-10 |
| | #07 | Print publication | *Starte og drive en frivillig organisasjon* | Final deliverable, solution report | 2018-10-28 / 2021-10 |
| Floke Innovation programme | #08 | Digital presentation | *Floke 2016-2021 - Feedback rapport* | Programme report, summary of learnings | 2021-01-12 / 2021-10 |
| | #09 | Digital document | *Byggfloken 2.0_kunnskapsgrunnlag* | Knowledge fundament for Floke project | 2020-06-10 / 2021-10 |
| | #10 | Digital presentation | *Byggfloken2.0_introduksjon* | Main document including background, description and scope | 2021-08-01 / 2021-10 |
| | #11 | Digital document | *Prosjektplan Bygg 2.0* | Project and process plan | 2021-08-05 / 2021-10 |
| | #12 | Print publication | *Byggfloken 2.0 Nøkkelinnsikter* | Report, project findings | 2021-09-29 / 2022-02 |
| | #13 | Print publication | *Byggfloken 2.0 konseptportefølje* | Final deliverable, process and evaluation report | 2022-03-17 / 2022-03-25 |
| | #14 | Digital presentation | *Byggfloken 2.0 Tek Pluss Porteføljelansering* | Final deliverable, solution presentation | 2022-03-17 / 2022-03-25 |

**Figure A1.** A complete list of documents used in the study.

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
