# Peer review of "Towards Systemic Innovation Programmes for Sustainability Transitions: A Comparative Study of Two Design-Led Cases"

_sustainability, doi:10.3390/su151310182_

Round 1

Reviewer 1 Report

This study investigates the structural organisation and theoretical lineage of two systemic innovations through the perspective of systemic design and transition management. Two systemic initiatives in Norway were analysed and compared - The StimuLab programme and Floke initiative.

It is interesting and well designed study.

I would suggest presenting also one additional aspect in the results section, namely the maturity of the system, showing not only the learning process of the initiatives but also how the initiatives mature overall. For more information please see: 

Abideen, Mr & Fiap, Tetlay & Philip, Prof & Freng, John & Mr, Copyright & Tetlay, Abideen & John, Phil. (2023). Clarifying the Concepts of System Maturity, System Readiness and Capability Readiness through Case Studies, or

Ryan Gove, Joe Uzdzinski,A Performance-Based System Maturity Assessment Framework, Procedia Computer Science, Volume 16, 2013, Pages 688-697, https://doi.org/10.1016/j.procs.2013.01.072.

Author Response

Dear Reviewer,

I truly appreciate your suggestion on including the perspective of system maturity, and the accompanying references. 
Indeed, the two cases studied are examples of highly persistent and enduring systems. I will do my best to incorporate this aspect in my revised manuscript.

Update:
I have added a brief mentioning of maturity in a separate section: 

4.3 General consideration of the cases, relating to transferability and applicability

Reviewer 2 Report

The present article proposal is characterized by an eminently descriptive approach without integrating a definite practical component to validate the proposed research topic.

I appreciate in favorable terms the foray made into the current state of knowledge of the consecrated aspect and the list of bibliographic references mentioned.

On the other hand, regarding the application part of the article proposal, I consider it unrealized and in this sense I recommend the development of a quantitative model to be validated later by testing with the help of some databases.

Therefore, taking into account the mentioned, I think that this article proposal deserves to be redone for its quantitative application part in order to receive the approval to be published later.

Author Response

Dear reviewer, 

I appreciate your argument for a qualitative element to be added and as such, improve its applicability. I have attached a more thorough response attached as a document (pdf), in which I elaborate on the choices made in terms of qualitative, case study approach and scope of the article.

Reviewer 3 Report

IT IS NICE TO HAVE TREMONDOUS DATA ANALYSIS TO HAVE SUCH A WONDERFUL COCLUSIONS.

Author Response

Dear reviewer,

I am truly thankful for your appreciation of this article.

Reviewer 4 Report

Somewhere in the discussion section the author should mention about the applicability of Stimulab and Floke for improving innovation/sustainability in rural areas since technical intervention in rural areas is a hot research topic. This manuscript should be accepted after minor revision. 

Author Response

Dear reviewer, 

I appreciate your suggestion for including perspective on rural contexts. It is indeed problematic that studies in Sustainability Transitions are largely limited to specific contexts (Global North, urban areas etc.)

Can you advice on possible reference that I could look into for such additional perspectives?

Update:
I have added a short mentioning of applicability in a separate section: 

4.3 General consideration of the cases, relating to transferability and applicability

Reviewer 5 Report

This is a well-written paper with a relevant focus. The starting sections of the paper are a bit too dense and not very clearly structured. This will require more editing for readability. The analysis sections of the case studies are well written and informative. I enjoyed the analysis and discussion. At the end a short section could be useful to summarize findings as the authors tend to write in a wordy, lengthy style that will be difficult for a broad audience to quickly scan for key insights and contributions.

Persistent small spelling, punctuation and grammar errors should be edited throughout. The author should prioritize simplicity and clarity in their writing as opposed to sounding elaborate at the expense of readability. A characteristic example in line 270: "Consequently - systemic practices that are situated, social and relational are of key foci."

Figure 1: A spelling mistake in "Sustainability Transition"

uppercase/lowercase consistency checking, especially in proper names, is required throughout all figures and all text

line 68ff - double check sentence

line 141 "Designerly approaches have evolved a distinct propensity towards human-centred and constructivist approaches [38] and thus exhibited their efficacy in complex, problem- solving environments through their capacity to engage with dynamics of both human and  built phenomena in context."

> Please clarify this sentence. I am not sure design can be equated with problem solving (that describes engineering). Why suddenly the focus on "built phenomena"?

line155

> please edit the sentence for clarity

Figure 2 is relevant and interesting but features a headline 2 in the middle of the diagram?

The image is explained in the following paragraphs but there are too many lists which distract readers, can this section be edited and integrated for readability?

It is really difficult for readers to keep track of TM, SIP etc especially since they seem to be amalgamated in and after Section 2. Some more clarity and distinction would be good. I am not sure the abbreviations help.

The last paragraph of the Methods section could be omitted.

line 281: "were deemed" - please be specific and give particular reasons. You selected them.

line 289: "ii" should be "iii"

The author refers to "solving challenges" frequently. This sounds like an engineering approach rather than design approach to me. In sustainability transitions should challenges not first of all be selected and defined collectively, and due to their intricate and systemic nature be "addressed" rather than "solved"?

Figure 4 is not very clear. Maybe it is not necessary to express everything with one diagram, and it would be good to focus on specific aspects of the framework at one time? In the end it is most important to have a clear overview or unit of analysis.

The headline of Section 5.6 is obviously repeating 5.5

Section 5.7 is now titled 5.6

Author Response

I appreciate your very thorough and considerate review- and I agree to most of your suggestions.

Please see the attached pdf for my point-by-point response on the main issues.

Round 2

Reviewer 2 Report

It states that for the application part of the article proposal, there is no empirical validation based on databases and quantitative models to test and validate the research undertaken.

Therefore, I recommend rethinking the application part by identifying quantitative models and databases to test and validate those models.

Author Response

 I fully appreciate your view that this study is unrealized in terms of application, and indeed I believe that follow-up research could venture into quantifiable model development. However, I would like to re-emphasise the circumstances for this study: The type of ambitious cross-sectoral, multi-actor systemic initiatives are currently a very rare breed of case examples.

Furthermore- the limited literature on the programmatic element argued for a qualitative approach in which the case study was chosen more in the function as a set of critical cases, constituting a powerful example (Siggelkow, 2007), with the intent to understand “how” and “why” (Yin, 2003).

I do believe there is a possible follow-up study that could advance my initial ideas related to the structural importance of systemic innovation (programmes), moving the findings from early generalizable characteristics towards typology and even quantifiable studies.

However, I believe this study then acts as a necessary forerunner that has identified key aspects and characteristics that could form the basis for a new article along the lines of your suggestions.

Round 3

Reviewer 2 Report

Unfortunately, I note the lack of a practical application study through which to validate the theories and theoretical developments of this article proposal.

Even if this article proposal is based on an appreciable literature, the prevailing narrative style and the lack of any empirical validation make me recommend major interventions on this proposal.